# Diversity, Pattern, and Environmental Drivers of Climbing Plants in China

**DOI:** 10.3390/plants14213281

**Published:** 2025-10-27

**Authors:** Haoran Wang, Guangfu Zhang

**Affiliations:** Key Laboratory of Biodiversity and Biotechnology, School of Life Sciences, Nanjing Normal University, Nanjing 210023, China; 231202005@njnu.edu.cn

**Keywords:** climbing plants, species richness, distribution pattern, environmental variables, China

## Abstract

As a distinct plant functional group, climbers critically sustain ecosystem structure and function globally. However, little is known about those in China. Here, we examine the diversity and distribution of Chinese climbers at a regional scale. First, climbing species data were collected. Then, Pearson correlations were conducted to assess relationships between environmental variables and climber species richness. Also, variation partitioning was used to reveal the pure and shared effects of four explanatory variable groups on species richness. A total of 3485 climber species (551 genera, 105 families) were recorded in China. Woody lianas dominated the climbing flora (64.73% of species) relative to herbaceous vines; twining represented the predominant mechanism (1829 species, 52.48%) relative to the others. Chinese climbers largely presented a pattern of species richness that decreased from south to north in China. Moreover, endemic and threatened climbers exhibited strong distributional congruence with all climbers. Additionally, four predictor groups (temperature, precipitation, geography, human impact) were found to jointly account for over 70% of species density variance across different climber types through variation partitioning, with precipitation’s pure effect dominating. Thus, Chinese climbers exhibit high diversity and an uneven distribution, primarily driven by precipitation. This study also provides a valuable reference on climbers at the regional scale for future studies.

## 1. Introduction

Climbing plants represent a unique category in nature since their stems are unable to stand upright and lack sufficient mechanical strength to support their own weight [1]. During their growth, they typically adopt climbing methods such as twining or sprawling, or rely on specialized organs like tendrils, haustoria, or adventitious roots to cling to other supports so as to enable upward growth [2]. Generally, plants with climbing habits are described as “climbing plants” or “climbers”, and woody climbers and herbaceous climbers are referred to as “lianas” and “vines”, respectively [3]. As early as 1875, Darwin observed the oscillatory movements of climbers’ stems and tendrils, which laid the foundation for subsequent studies on their behavior, and he further classified climbing plants into four categories according to attachment modes: twining, hook and leaf-bearers, tendril-bearers, and root climbers [4]. Recent studies indicate that over one-third of the families of seed plants worldwide contain climbing plants, suggesting that the climbing ability of plants has evolved multiple times [5]. Climbers typically attach themselves to neighboring plants to grow vertically so that they can access more light resources and growth spaces; moreover they employ diverse climbing methods to promote their abundance and survival [6,7]. Climbing plants often grow faster than trees because of their strong adaptability, which is due to their flexible stems, rapid growth rates, and extensive root systems, as well as preferences for light [8]. In fact, climbers play a key role in forest ecosystems and are indispensable components of the world’s forest vegetation, particularly in tropical and subtropical forests [1,9]. For example, many woody lianas have the functions of stabilizing forest structure and enhancing species richness and biomass [10]. Additionally, some climbers are widely utilized as food [11], as medicine [12], and in landscaping [13].

The species number of climbing plants is globally estimated to exceed 25,000, with most species concentrated in a few families or genera. For instance, Apocynaceae and Fabaceae are the two largest families containing climbers worldwide, each with over 2000 species [14]. Current studies on climbing plants have primarily focused on climbing mechanisms [15], functional traits [16], and phylogenetic analyses [17]. However, climbers have received significantly less attention in terms of their diversity and distribution than other plant groups. Global climber diversity exhibits marked spatial and temporal heterogeneity [18]. Lianas account for 35% of the whole woody plant species from tropical forests, with such a proportion reaching the peak compared to those from other biomes, in which lianas are intrinsically linked to closed-canopy forests in distribution [1]. Furthermore, climbing plants are more diverse and abundant in tropical forests than in temperate forests, and they exhibit an intermediate level in subtropical forests [19,20]. Climbers, similar to other life forms, are influenced by various factors including altitude, latitude, climate, and human impacts in terms of abundance and diversity [21,22], leading to their geographical disparities between and within regions. Notably, due to their wide and long vessels, climbing plants are more susceptible to embolism caused by cold or drought compared to most plants with narrow and short vessels [23,24]. Consequently, most current studies have focused on tropical climbers, with few on subtropical and temperate climbers. In addition, it is a big challenge to identify climbing plants during field surveys compared with other plants since climbing plants have difficulties in forming independent communities and have no fixed layer positions [18]. The great majority of woody lianas are located in the canopy layer during late forest succession, and therefore, they are often overlooked [25].

Studies on climber diversity and distribution have been conducted across global, regional, and local scales, yet significant gaps persist. At the global scale, studies such as those by [1,14] have documented climbing plants across 133 to 169 families, while DeWalt et al., using the Global Liana Database, revealed notable biogeographical disparities—climber density in Neotropical and African forests was nearly double that in Asia [26]. Hu and Li further identified Southeast Asia, South China, South Asia, and the Himalayas as four hotspots with over 1000 climbing plant species each [14], though global data have considerable deficiencies in climbers. At the regional scale, studies in temperate East Asia [27], South America [28], Mexico [29], India [30], and China [31,32] have enriched climber species inventories and taken to exploring environmental drivers, yet these efforts remain fragmented and understudied. In contrast, local-scale studies are abundant, including those employing plot sampling in Panama [33], Malaysia [34], and New Caledonia [35] and those conducting local investigations in some localities of China [36,37,38] and Brazil [39]. However, most local studies lack integrative or statistical analysis.

Nowadays, numerous local-scale studies have focused on compiling plant checklists, describing species composition, and classifying climbing methods. Although such studies are feasible and straightforward in practice, they have apparent drawbacks. For one thing, climbing plants are mainly distributed in warm and humid zones [40]. Studies confined to local scales, however, suffer from limited geographical range. Accordingly, it is difficult to reveal the causes of the distribution of climbers at this scale. For another thing, the restricted comparability across such localized studies hinders the extrapolation of findings to other sites. Conversely, there are relatively more species of climbers in the scope of medium-scale (i.e., regional scale), covering more distinctive environmental gradients, than in the scope of local scale. Indeed, the global-scale studies on climbers may merely draw general or common-sense conclusions. Moreover, it is difficult to identify the driving factors of climbing plants due to data deficiency. Therefore, we think that it is appropriate to study the diversity, distribution, and drivers of climbing plants at the regional scale.

China is characterized by its vast territory, diverse climates, and complex landforms, making it one of the countries with the richest plant diversity in the world [41,42]. Ranking third globally in terms of territorial area, China spans 5200 km from east to west and covers over 50° of latitude. It encompasses five climatic zones from south to north: tropical, subtropical, warm temperate, temperate, and cold temperate. The terrain is higher in the west and lower in the east, presenting a distinct three-step distribution, with altitudes of the first step reaching over 4000 m [43,44]. In recent decades, China has vigorously conducted biodiversity surveys and conservation efforts, with the *Flora of China* and local floras being published and continuously updated, thereby accumulating abundant species data. For instance, China annually releases an updated *Catalogue of Life China*. There are approximately 3000 climbing plant species in China [31], accounting for about 7.20% of the country’s higher plant species (41,687) [45]. However, studies on climbers remain relatively scarce. The white paper on “*Biodiversity Conservation in China*” states that, on average, nearly 200 new plant species have been discovered annually in China over the past decade, some of which belong to climbing plants. In the past several decades, new species and records of climbers have been continuously discovered in China. One example is the genus *Isotrema* from the family Aristolochiaceae. Integrative morphological and molecular evidence supports the segregation of *Isotrema* from *Aristolochia* at the generic rank. Zhu et al. [46] reported that there were 59 species belonging to *Isotrema* in China. Our latest statistics show that there are 78 species of this genus in China at present. Another example is the genus *Cheniella* from the family Fabaceae. Integrative morphological, molecular, and biogeographic evidence supports the segregation of some *Bauhinia* species into the new genus *Cheniella*. More recently, three new species from *Cheniella* have been recognized: *Cheniella hechiensis* S.R. Gu, T.Y. Tu & D.X. Zhang, *Ch. Pubicarpa* S.R. Gu, T.Y. Tu & D.X. Zhang, and *Ch. longistaminea* S.R. Gu, T.Y. Tu & D.X. Zhang [47]. In summary, China can serve as an ideal research platform for analyzing the distribution patterns and environmental drivers of climbing plants because of its comprehensive advantages in species diversity, habitat heterogeneity, and data completeness.

Our literature analysis reveals three major limitations in regional-scale studies of Chinese climbing plants. (1) Outdated taxonomic data and records (i.e., rejected nomenclature, taxonomic revision, imprecise locality). For instance, Hu et al. [31] relied exclusively on *Flora Reipublicae Popularis Sinicae* (1959–2004) for compiling a checklist of Chinese climbers. Subsequently, extensive field investigations were conducted and numerous taxonomic revisions were made regarding China’s vascular plants. (2) Not distinguishing wild from cultivated species. Typically, for a plant species, its introduced ranges do not accurately reflect its natural distribution patterns and suitable environmental conditions. (3) Lack of distribution pattern of all climbers in China. Hu et al. [32] only selected 82 floras from different provinces, representing different climate zones in China, as representative climber plots, but some provinces (e.g., Taiwan Province) were missing in their study.

Here, we first compiled the most comprehensive checklist of Chinese climbing plants, and further examined their species richness and distribution pattern. Specifically, we aim to: (1) Analyze the floristic composition, life form, and climbing method of these climbers in China; (2) Characterize the spatial pattern of entire, endemic, threatened, and invasive climbing plants across China; (3) Identify the primary environmental determinants for Chinese climbing plants and each life form. The main objective of this study is to reveal the diversity characteristics and spatial distribution patterns of climbing plants in China (hereafter referred to as “CPC”), explore their driving environmental factors, and provide a reference for regional-scale climbing plant research.

## 2. Results

### 2.1. Floristic Composition of Climbing Plants in China

A total of 3485 climbing plant species (including infraspecific taxa) belonging to 551 genera and 105 families were recorded in China. Among them, 3398 species were wild, while only 87 species were cultivated. There were 1880 climbing plant species endemic to China, accounting for 53.95% of CPC (Appendix A). CPC comprised three taxa: lycophytes and ferns, gymnosperms, and angiosperms. Lycophytes and ferns had 3 families, 3 genera, and 11 species; gymnosperms had 1 family, 1 genus, and 10 species; angiosperms had 101 families, 547 genera, and 3464 species (Table 1). Overall, angiosperms dominated CPC at the family, genus, and species levels, accounting for 96.19%, 99.28%, and 99.39% of the total, respectively.

Among the 3485 climbing plant species in China, the family with the highest number of species was Fabaceae, with 352 species, accounting for 10.10% of the total. Apocynaceae ranked second with 338 species (9.70%), followed by Ranunculaceae with 229 species (6.57%).

Based on the number of species within each family, the families of CPC could be categorized into four groups (Table 2). Firstly, there were nine large families with more than 100 species, accounting for only 8.57% of the total families of CPC. However, these nine families collectively comprised 242 genera (43.92% of the total genera) and 1893 species (54.32% of the total species), indicating a significant dominance of large families in Chinese climbing plants. Secondly, there were 56 medium families containing 6–100 climbing plant species, totaling 1515 species (43.47% of the total). Thirdly, there were 19 oligospecific families (2 ≤ *N* ≤ 5) with 34 genera and 56 species, and 21 monotypic families (*N* = 1) with 21 genera and 21 species. Oligospecific and monotypic families collectively accounted for 40 families, 55 genera, and 77 species (38.10% of the total families), highlighting a large disparity in the number of species among different families.

### 2.2. Life Forms and Climbing Methods of Climbing Plants in China

There were three life forms of CPC. Among them, evergreen woody lianas were the most abundant, with 1362 species (39.08% of the total); herbaceous vines ranked second with 1229 species (35.27%); and deciduous woody lianas were the fewest, with only 894 species (25.65% of the total) (Table 3). Overall, woody lianas accounted for 64.73% of the total climbing plant species, nearly twice as many as herbaceous vines.

CPC employed four climbing methods: twining, tendrillar, adhesive, and sprawling. Twining climbers were the most prevalent, with 1829 species (52.48% of the total), followed by sprawling climbers (877 species; 25.17%), tendrillar climbers (565 species; 16.21%), and adhesive climbers (214 species; 6.14%) (Table 3). Active climbers (twining and tendrillar climbers) accounted for 68.69% of the total climbing plant species, and passive climbers (adhesive and sprawling climbers) accounted for the remaining 31.31%.

Climbing mechanisms differed significantly in composition across climbing plants’ life forms. For evergreen woody lianas, twining climbers (712 species; 52.28%) were the most abundant, followed by sprawling climbers (373 species; 27.39%), tendrillar climbers (148 species; 10.87%), and adhesive climbers (129 species; 9.47%). Deciduous woody lianas were similar to those of evergreen woody lianas in composition and proportion of climbing method. For herbaceous vines, twining climbers (686 species; 55.82%) were also the most abundant, and adhesive climbers (46 species; 3.74%) were the fewest. However, the number of tendrillar climbers (269 species; 21.89%) slightly exceeded that of sprawling climbers (228 species; 18.55%). Overall, twining was the predominant climbing method across different life forms of CPC.

### 2.3. Spatial Distribution Patterns of Climbing Plant Diversity at the Family and Genus Levels in China

According to the F-index value, the top three provinces were Yunnan (62.42), Guangxi (53.03), and Hainan (50.51) in China, with each of them greater than 50. The three provinces with the lowest F-index value were Xinjiang (4.79), Nei Mongol (4.14), and Ningxia (3.81), each less than 5. The average F-index value for the 28 geographical units was 26.24 (Figure 1a). Southwestern China (Yunnan: 62.42; Guizhou: 45.34; Sichuan: 38.20; Xizang: 28.83), Southern China (Guangxi: 53.03; Hainan: 50.51; Guangdong: 48.93), and Eastern China (Taiwan: 44.63; Fujian: 36.25; Jiangxi: 32.41; Zhejiang: 31.68; Anhui: 25.52; Jiangsu: 20.06; Shandong: 12.56) exhibited higher F-index values, with most provinces in these regions having F-index values above the average. Conversely, Northwestern China (Shaanxi: 24.01; Gansu: 20.31; Qinghai: 5.45; Xinjiang: 4.79; Ningxia: 3.81), Northern China (Shanxi: 12.77; Hebei: 12.21; Nei Mongol: 4.14), and Northeastern China (Liaoning: 11.33; Jilin: 10.25; Heilongjiang: 8.52) exhibited lower F-index values, with each unit less than the average. Overall, these climbing plants showcased a pattern of familial diversity decreasing from south to north across China.

Similarly, in terms of the G-index, the top three provinces were Guangdong (5.69), Hainan (5.68), and Yunnan (5.63). The province with the lowest G-index was Ningxia (3.03), followed by Nei Mongol, Qinghai, and Xinjiang, all with G-index values of 3.04. The average G-index value for the 28 geographical units was 4.27 (Figure 1b). Overall, the G-index exhibited similar distribution pattern to the F-index in China.

### 2.4. Spatial Distribution Patterns of Climbing Plant Species Richness at the Provincial Scale in China

Figure 2 illustrates the spatial distribution patterns of entire, endemic, threatened, and invasive climbing plant species in China. For the entire climbers, Yunnan had the highest number of species (1942), followed by Guangxi (1263) and Guangdong (907). Additionally, there were six provinces with climbing plant species richness less than 100: Jilin (95), Heilongjiang (86), Nei Mongol (72), Xinjiang (72), Qinghai (61), and Ningxia (48), which were located in Northeastern, Northern, and Northwestern China. Overall, the species richness of climbers was the highest in Southwestern China, but comparatively low in Northeastern, Northwestern, and Northern China (Figure 2a). CPC was highly uneven in distribution, largely showing a pattern of species richness decreasing from south to north.

Endemic climbing plant species were mainly concentrated in Southwestern and Southern China, with Yunnan having the greatest number of species (892), accounting for 47.44% of the total endemic climbers in China (1880). Guangxi (543) and Sichuan (520) ranked second and third, respectively. Heilongjiang had the fewest endemic climbers (9), followed by Nei Mongol (10), Xinjiang (11), and Ningxia (11), which were also located in Northwestern, Northeastern, and Northern China (Figure 2b). Overall, endemic climbers in China showed a similar distribution pattern to that of total climbing plant species.

For threatened climbers, Yunnan had the highest number of species (322), followed by Guangxi (160) and Hainan (93). The three provinces were located in Southwestern and Southern China. There were 12 provinces with fewer than 10 threatened climbing plant species, and most of them were located in Northwestern, Northeastern, and Northern China. The three provinces with the fewest threatened climbers were Qinghai (2), Nei Mongol (1), and Ningxia (0) (Figure 2c). Therefore, threatened climbers in China still showed a pattern of diversity decreasing from south to north in distribution. Additionally, for invasive climbers, there were only 18 species (0.52% of the total climbers in China), much fewer than the other categories. Most provinces in China only had 1–4 invasive climbing plant species (Figure 2d), most of which were concentrated in the southeastern coastal provinces of China (such as Fujian, Guangdong, Hainan and Taiwan).

### 2.5. Correlations Among the Entire, Endemic, Threatened, and Invasive Climbing Plant Species Density in China

The correlation coefficients of climber density between the entire species and the endemic/threatened species were greater than 0.9, indicating a strong correlation between them. The correlation coefficient between the whole and the invasive species was greater than 0.7, indicating a relatively high correlation between them. Additionally, the correlation coefficient between the endemic and the threatened species was 0.912, suggesting a significant correlation between them (Table 4).

### 2.6. Determinative Environmental Factors Influencing the Density of Climbing Plants in China

Thirteen of the 23 environmental variables were significantly correlated with CPC density, including eight temperature-related, four precipitation-related, and latitude among spatial predictors. Among them, Bio7 (temperature annual range) showed the strongest correlation with climbing plant density (*r* = −0.876), followed by latitude (*r* = −0.864) and Bio4 (temperature seasonality) (*r* = −0.862) (Table 5). Apart from the significant negative correlation between latitude and climber density, the other two variables (longitude and altitude) of geographical location presented relatively weak correlations with climber density. Additionally, the correlation between human impact and climbing plant density was also weak (*r* = 0.227). Collectively, the density of CPC was closely related to temperature and precipitation.

The density–environment relationship for each life form paralleled that of the entire climbing plants in China. For instance, among the 11 temperature-related variables (Bio1-Bio11), Bio7 consistently showed the strongest correlation with the density of each life form, with each value exceeding 0.8. Conversely, the human impact variable was weakly correlated for each life form (i.e., |*r*| < 0.33).

Due to the strong collinearity among climatic variables, cluster analysis was employed to group the 19 climatic variables into eight clusters. Only one climatic variable selected from each cluster in light of the maximum correlation coefficient (|*r*|) within the cluster was retained for subsequent variation partitioning analysis. The final eight selected climate variables were shown in Appendix A.

For the entire climbers, the total percentage of variation explained by the four explanatory variable groups that we selected was 74% (Figure 3). According to the variation partitioning results, the pure effect of precipitation contributed the most to explaining the total species distribution pattern (X2|X1 + X3 + X4, R^2^ = 0.44). Regarding the pure effects of other variable groups, temperature and geographical location also explained a considerable proportion of the total variation (X1|X2 + X3 + X4, R^2^ = 0.12; X3|X1 + X2 + X4, R^2^ = 0.28). Notably, the shared effect of all four variable groups and of temperature and precipitation exactly accounted for the same proportion of the explained variation (X1 + X2 + X3 + X4, R^2^ = 0.08; X1 + X2, R^2^ = 0.08).

The four groups of explanatory variables accounted for more than 75% of the variation in each life form (evergreen woody lianas, deciduous woody lianas, and herbaceous vines), indicating that the selected environmental variables could effectively explain the variations in climbing plant species density. According to the variation partitioning results, the pure effect of precipitation accounted for the highest proportion of the variation in the climber densities of three life forms (X2|X1 + X3 + X4, R^2^ = 0.50, 0.45, 0.49). The shared effect of all four environmental variable groups and the shared effect of temperature and precipitation also explained a portion of the variation for each life form, respectively (X1 + X2 + X3 + X4, R^2^ = 0.09, 0.08, 0.08; X1 + X2, R^2^ = 0.06, 0.08, 0.09). In terms of the pure effects of other environmental variable groups, temperature and geographical location explained greater proportions of the variation in the climber densities of three life forms relative to shared effects, respectively (X1|X2 + X3 + X4, R^2^ = 0.13, 0.12, 0.07; X3|X1 + X2 + X4, R^2^ = 0.24, 0.31, 0.15).

## 3. Discussion

### 3.1. Diversity Characteristics of Climbing Plants in China

Our analysis indicates that China harbors a total of 3485 climbing plant species (including infraspecific taxa) belonging to 551 genera and 105 families, which is more than that reported by Hu et al. [31] (i.e., 3073 species from 85 families and 409 genera). Firstly, they compiled the checklist of Chinese climbing plants, exclusively based on *Flora Reipublicae Popularis Sinicae* Volumes 7—80 (1961–2002). On the contrary, we completed an updated checklist by collating *Flora of China* Volumes 1—25 (1988–2013) and the *Catalogue of Life China (2025 Annual Checklist)*, together with relevant survey data. Moreover, we have supplemented climbing plant species over the past several decades, particularly for new species or records in China. In addition, we adopted the latest classification system. For example, we referred to APG IV rather than Engler system for angiosperms.

Generally, climbing plants are considered as one of the important components of the world’s flora. Currently, they account for 8.33% of known higher plants (ca. 300,000 species) in the world [48]. The proportion of climbing plants in China is 8.36%, slightly higher than global percentage. The species number of climbing plants in China is greater than that in India, where climbers (2624 species from 104 families and 585 genera) constitute 15.44% of all higher plants [30]. Meanwhile, this number is higher than that in Australia, where climbers (834 species from 91 families and 286 genera) constitute approximately 4% of all higher plants [49]. Therefore, China has relatively rich species diversity of climbing plants.

Endemicity is one of the most important characteristics of flora at a regional scale. Our results have shown that there are 1880 climbing species endemic to China, accounting for 53.95% of the total climbers in China. This is probably related to the fact that China has rich plant diversity. China has as many as 41,687 higher plant species, of which 21,457 (51.47%) are endemic [45]. In other words, over half of the higher plants in China are endemic. The proportion of endemic climbers in China (53.95%) is slightly higher than that of endemic higher plants (51.47%). This high endemism may be ascribed to the vast land area, various climate zones, and diverse habitat heterogeneity in China. These will help to create a wide range of habitats and may well promote local species differentiation, resulting in the emergence of endemic climbers [50,51,52]. Additionally, climbing flora of China exhibits pronounced dominant families. Nine large families collectively represent 54.32% of entire Chinese climbers (Table 2), reflecting a significant disparity in species richness among families. This may be potentially explained by the concentration of climbers in the south of China, where abundant hydrothermal conditions favor their growth and reproduction.

As a particular type of life form, climbers play a significant role in the vegetation of a region [24]. In the current study, we identify three distinctive categories in terms of life forms for CPC, predominated by evergreen woody lianas. Woody lianas in China are about twice that of herbaceous vines in species number. This is most likely due to most climbers in China being distributed in tropical and subtropical regions like Southern and Southwestern China (Figure 2a). Furthermore, Chinese climbers have four climbing methods, with the great majority employing the twining method. It is reported that twining climbers account for the largest proportion in terms of species richness and abundance in many tropical regions around the world [53]. The possible reason is that twining enables climbing plants to utilize support structures with the widest range of diameters relative to other climbing mechanisms [54].

In conclusion, Chinese climbing flora has a high level of diversity, mainly characterized by rich species, high endemism, various life forms, and diverse climbing methods.

### 3.2. Spatial Distribution Pattern of Climbers in China

Taking China as a case study, we examined the spatial distribution patterns of climbing plants using four metrics: diversity indices at the family and genus levels (i.e., F-index in Figure 1a, G-index in Figure 1b), species richness (Figure 2a), and species density (Appendix A). The results reveal an uneven spatial distribution of climbers across China, with diversity gradually decreasing from south to north. Our results show that over half of climbers in China are endemic (Appendix A), indicating a highly significant positive correlation between entire and endemic climber density (*r* = 0.950; *p* < 0.01) (Table 4). In other words, the provinces with rich climber diversity also harbor comparatively abundant Chinese endemic climbers. Our results also indicate a highly significant positive correlation between entire and threatened climber density (*r* = 0.963; *p* < 0.01) (Table 4). Thus, this reflects a congruent distribution pattern between Chinese endemic/threatened and entire climbers. A strong positive correlation (*r* = 0.912, *p* < 0.01) was observed to exist between threatened and endemic climbers in China (Table 4). This is primarily because most endemic species have restricted distribution ranges and are vulnerable to man-made disturbances, rendering them threatened in the wild. Given that the greater majority of these endemic climbers are distributed in Southwestern (e.g., Sichuan, Yunnan) and Southern China (e.g., Guangdong, Guangxi), such areas could be designated as priority areas in conservation (Figure 2b). Unlike endemic or threatened climbers, invasive climbers present low correlation coefficients with the other three categories in species density, all reaching the significance level (*p* < 0.01) (Table 4). Most of them are found in provinces like Fujian, Guangdong, Hainan and Taiwan within southeastern coastal regions (Figure 2d), resulting from their developed international trade, frequent human mobility, and well-established transportation networks. We propose that more attention should be paid to invasive climbing plants for these coastal areas of China in the future.

Such a distribution pattern of climbers in China is similar to that in Mexico, where climbing plants are also unevenly distributed across 32 states, with the highest species richness concentrated in the tropical southeastern region [29]. It is worth noting that Mexico is complex and diverse in climate, and thereby has a rich flora of climbing plants, reaching 754 species in total. However, Mexico is only one-fifth of China in territory area, making it fall into the category of local scale. Therefore, this study is the first to explicitly determine the diversity pattern of climbing plants in China at the regional scale.

The spatial distribution of plants is shaped by both biotic and abiotic factors [55]. Firstly, the distribution patterns of climbers in China aligns with those of Chinese woody plants [56] and higher plants [45]. Climbing plants often require support structure (i.e., trees) during their growth process, particularly for twining climbers. Our results show that more than half of Chinese climbers employ twining as a climbing method (Table 3). Therefore, the species richness of climbers in a geographical unit is closely associated with the overall plant diversity, which is similar to the positive correlation observed between the diversity of parasitic plants and seed plant richness in China [52].

Secondly, climate is a key factor influencing the distribution of plants at the regional scale [57]. At the provincial level, Yunnan (located in Southwestern China) exhibits the highest species richness of climbing plants, while Ningxia (in Northwestern China) has the lowest (Figure 2a), with the former being 40 times greater than the latter. Among China’s seven geographical regions, Southwestern China (1042, average species richness across all provinces within the region, the same below) and Southern China (940) show comparatively high species richness, whereas Northwestern (148), Northern (112), and Northeastern China (95) have significantly low richness. Thus, the spatial distribution of Chinese climbers, in provincial units or geographical regions, presents a pronounced pattern of decreasing from southern towards northern across the country, mainly resulting from local climate and hydrothermal conditions. China spans five climatic zones from south to north. Overall, the southern part of China mainly has a subtropical climate, while the northern part has a temperate monsoon climate. Compared to the north of China, there are more suitable water and heat conditions in the south of China, which is more conducive to the distribution and growth of climbers. Taking Yunnan in Southwestern China as an example, its outstanding climbing plant diversity (i.e., 1942 species) can be primarily attributed to diverse climates. Yunnan Province usually experiences average July temperatures between 20–24 °C and January averages between 7–13 °C, and it has a long frost-free period, with the southern region over 300 days [58]. Located on the country’s second topographic step, Yunnan is affected by both the south subtropical and East Asian monsoons, resulting in humid and diverse climate [59,60].

In contrast, Northeastern China experiences a temperate monsoon climate, spanning middle and cold temperate zones [27,61]. Its mean annual temperature ranges only from −4.2 to 12.9 °C [62]. Due to prolonged aridity and cold, climbing plants in temperate forests may face additional constraints: they tend to shed leaves earlier than trees to avoid potential vascular damage from unseasonal early frosts [19]. These environmental stresses lead to the relatively low diversity of climbers in this region. Unlike the warm and humid climate in the south of China, Northwestern China has severe arid conditions, primarily resulting from the blocking effect of mountain ranges such as the Tibetan Plateau, Taihang Mountains, Yanshan Mountains, Qinling Mountains, and Greater Khingan Mountains on monsoon circulation. These barriers effectively prevent the northward and westward advancement of moist monsoons from southern and eastern China, leading to dry and low-rainfall climate [63,64].

Thirdly, geographical conditions also play a significant role in shaping the distribution patterns of climbers [65]. There are significant differences in geographical conditions between the north and the south of China, presenting obvious complexity and diversity. The Qinling Mountains–Huaihe River line generally serves as the boundary between the temperate monsoon climate to the north and the subtropical monsoon climate to the south [66]. Moreover, our analysis indicates that latitude has a significant negative correlation with the adjusted species density of climbers (Table 5). In this case, latitude serves as a geographical location variable. Latitude determines solar radiation intensity: lower latitudes receive more radiation and higher temperatures, while higher latitudes receive less radiation and lower temperatures [67]. Consequently, Southwestern and Southern China, situated at lower latitudes, have higher climbing plant species richness compared to other geographical regions.

In summary, the spatial distribution of Chinese climbers presents a pattern of high in the south and low in the north across China, mainly resulting from the climatic factors.

### 3.3. Key Environmental Factors Influencing the Distribution Pattern of Climbers in China

Variance partitioning is generally considered an effective statistical tool for separating and quantifying the relative influences of environmental variables, thereby making key drivers more identifiable and interpretable [68]. In this study, we initially selected 23 environmental variables across four categories (i.e., temperature, precipitation, geographical location, and human impact) and calculated their Pearson correlation coefficients with climber density across China (Table 5). To reduce the multicollinearity among the 19 climate variables including temperature and precipitation, we performed cluster analysis to group them into eight clusters. Only one variable from each cluster was retained based on the correlation coefficient between climate variables and climber densities (Appendix A). We further applied variance partitioning to quantify the contributions of four explanatory variable groups including 12 factors to the variance in climber density of China. Similarly, this approach was also employed for the climbers of each life form. The results showed that: (1) explanatory variables of the four groups explained over 70% of the variance in species density of each type (i.e., entire climbers, evergreen woody lianas, deciduous woody lianas, and herbaceous vines). For instance, the four groups’ explanatory variables can explain almost 80% of the variation in the density of evergreen woody lianas. To our knowledge, this is the first to identify environmental drivers of climber distribution patterns in China with variance partitioning. Moreover, it can provide a methodological reference for corresponding regional-scale studies. (2) Among the four groups’ variables, the pure effect of precipitation accounted for the highest proportion, independently explaining over 40% of the density variation for each type. One example in point is evergreen woody lianas. The pure effect of precipitation can explain up to 50% of the variation in their density (Figure 3). Our result is consistent with the conclusion from Hu et al. [32], who contended that drought stress served as a major constraint on the distribution of climbers across 82 floras in China, and that the proportion of climbers decreased with the reduction of precipitation from south to north. Likewise, Parolari et al. [69] reported that the abundance and diversity of climbers in Panama increased with precipitation seasonality. Gallagher [49] also found that limited rainfall restricted the distribution of climbing plants in arid inland regions of Australia.

Such a high explanatory proportion of precipitation for the variation in climber density across China can be attributed to the following three aspects. First of all, our findings indicate that the pure effect of precipitation is the primary driver shaping the distribution patterns of Chinese climbers for different life forms (Figure 3). This is most likely due to the characteristics of climbing plants. Generally, for free-standing plants, their stems have to serve as supporting structure and transporting water; however, climbers employ various climbing methods, such as twining, sprawling, and adhering (i.e., root-climbers), to extend their stems or branches by ascending host trees, thus expanding their living space and obtaining light resources for photosynthesis [24,70]. Since climbing plants do not need to build upright stem structure, they reduce the carbon allocation to woody tissues and the cost of water transport from soil to canopy, resulting in significantly higher water transport efficiency [2,71]. As a special functional group, climber species usually exhibit large leaf size [49]. Larger leaves provide species with many strategic ecological advantages, including the ability to use a larger surface area for light capture during photosynthesis, especially the competitive advantage of monopolizing light resources in the canopy of forest communities. However, only under the condition of sufficient water can these advantages be achieved and maintained [72,73]. For instance, in terms of leaf area, climbing plants in temperate regions are on average four times smaller than those in tropical regions across Australia. Such a reduction likely reflects gradients in annual soil water availability, which is generally higher in tropical forests due to greater precipitation compared with temperate forests [49].

Secondly, like woody plant species or seed plants in China, climbers are primarily driven by climatic factors involving precipitation and temperature in distribution pattern [56,74]. This aligns with the hypothesis of energy and water availability [75]. Compared to other environmental variables, water availability exerts a stronger spatial influence on the species richness of seed plants across China [74]. Similarly, mean annual precipitation is identified as a key environmental determinant of woody plant diversity at the regional scale in China [56]. Generally, mean annual precipitation decreases from the southeastern coast to the northwestern inland, with the south of China receiving more rainfall than the north [76]. For example, Yunnan Province in Southwestern China has an average annual precipitation of 1379 mm, about 80% of which occurs during the rainy season (i.e., May to October) [77]. In contrast, Xinjiang in Northwestern China receives only about 150 mm annually [78], approximately one-tenth of Yunnan’s total. This national gradient in precipitation fits closely with the distribution pattern of climbing plants in China (Figure 2a). Moreover, in tropical regions such as Ghana, the density and diversity of woody lianas (relative to trees) are also strongly influenced by rainfall amount and seasonality [79].

Notably, other environmental factors such as temperature, topography, and anthropogenic impact have varying degrees of influence on precipitation, probably resulting from the interactions among these environmental factors [55]. For instance, as temperatures decrease, the capacity of the wide and long vessels, which are typical of woody lianas, to resist or recover from cavitation caused by freezing temperatures declines markedly, thereby imposing an adverse effect on their water transport efficiency [71].

### 3.4. Conservation Implications for Climbing Plants in China

Our findings have significant implications for the conservation of climbing plants in China. The “3030 target”, which is the core of Kunming-Montreal Global Biodiversity Framework, aims to protect 30% of the area of land, freshwater, and oceans by 2030 [80]. It is highly challenging to achieve this target. To meet it, China plans to expand protected areas to cover 18% of its land, utilizing other effective area-based conservation measures (OECMs) for the additional 12% [81]. OECMs (such as the mini natural reserves and the civil protected areas) can serve as a beneficial supplement to the nature reserve system or act as corridors to connect nature reserves, thereby promoting the formation of a more effective in situ biodiversity conservation network [82]. We believe that if the diversity of climbing plants (having rich species, high endemism, and various climbing mechanisms) can be taken into account when formulating OECMs, it may contribute to maintaining biodiversity level and ecological function. Particularly for the tropical region of southern China with abundant climbing plants (Figure 2), we propose integrating concerned climbers into the existing protection catalogue and strengthen climber habitat conservation or ecological corridor construction.

Our analysis reveals that 645 species are in the status of threat categorization (including EX, RE, CR, EN, VU, and NT), accounting for 18.51% of the total Chinese climbers (3485 species). Currently, the *List of National Key Protected Wild Plants* updated in 2021 encompasses 1043 species, including 106 lycophytes and ferns, 107 gymnosperms, and 830 angiosperms [83]. Among them, there are only 31 climbing plant species (2 first-class and 29 s-class national protected species), representing merely 2.97% of the total vascular plants in the list. We thus recommend incorporating additional climbing plant species into protected lists to bolster both legal safeguards and management frameworks.

## 4. Materials and Methods

### 4.1. Study Area

The sample units in this study are the provinces of China. China encompasses 34 provincial-level administrative regions, including provinces, autonomous regions, municipalities directly under the central government, and special administrative regions. Given the relatively small land areas and high levels of urbanization of the four municipalities (Beijing, Tianjin, Shanghai, and Chongqing) and the two special administrative regions (Hong Kong and Macao), they were merged with adjacent provinces [45,52]. Specifically, Beijing and Tianjin were incorporated into Hebei Province, Hong Kong and Macao were included within Guangdong Province, Chongqing and Shanghai were included in Sichuan and Jiangsu Province, respectively. As a result, 28 geographical units were ultimately created (Appendix A). Furthermore, based on China’s natural geography, these provinces are divided into seven geographical regions: Northwestern China (Gansu, Ningxia, Qinghai, Shaanxi, Xinjiang), Southwestern China (Guizhou, Sichuan, Xizang, Yunnan), Northeastern China (Heilongjiang, Jilin, Liaoning), Northern China (Hebei, Nei Mongol, Shanxi), Central China (Henan, Hubei, Hunan), Southern China (Guangdong, Guangxi, Hainan), and Eastern China (Anhui, Fujian, Jiangsu, Jiangxi, Shandong, Taiwan, Zhejiang) [45,64,84].

### 4.2. Data Collection and Organization of Climbing Plant Species

The species data of climbing plants were compiled based on the *Flora of China*, supplemented by the *Catalog of Life China* (2025 Annual Checklist), *Chinese Plant Names Index* (2000—2009, 2010—2017), and literature on new plant species published in recent years [85,86,87,88]. After sorting out the checklist of climbing plants, we arranged all families and genera according to the taxonomic system. Among them, lycophytes and ferns adopt the Pteridophyte Phylogeny Group (PPG) system, gymnosperms employ the Gymnosperm Phylogeny Group (GPG), and angiosperms utilize the Angiosperm Phylogeny Group IV (APG IV).

The life forms of climbing plants were categorized as evergreen woody lianas, deciduous woody lianas, and herbaceous vines, with reference to the dataset on the plant growth form and life form of vascular plants in China [89]. The endangered category of climbing plant species was determined according to the *Red List of China’s Biodiversity: Higher Plants Volume (2020)*. Climbing plants encompassing the categories of EX (Extinct), RE (Regionally Extinct), CR (Critically Endangered), EN (Endangered), VU (Vulnerable), and NT (Nearly Threatened) were considered as threatened species. Data on invasive climbing plant species were sourced from the species list of invasive alien plants and naturalized plants in China [90].

Following the classification method of Hu et al. [31], climbing plant species in China were classified into four climbing methods based on their utilization characteristics of potentially available support conditions: (1) Tendrillar climbers: These climbers possess specialized tendrils derived from stems, stipules, leaves, or pedicels, which may be branched or unbranched. They climb upward by winding their tendrils around external supports and are the groups with the highest climbing efficiency, such as *Smilax china* L. (Figure 4a). (2) Twining climbers: These climbing plants lack obvious specialized organs and accomplish their climbing through the twining mechanism induced by contacting between their stems, branches, or petioles and external supports. This represents the most prevalent, most primitive, and least efficient climbing method, such as *Lonicera japonica* Thunb. (Figure 4b). (3) Sprawling climbers: These species climb by sprawling and leaning on other plants with slender branches. They are often equipped with specialized organs such as thorns or hooks, or exhibit features like multi-branched, opposite phyllotaxy, or verticillation, which enhance climbing capabilities and improve stability, such as *Rosa cymosa* Tratt. (Figure 4c). (4) Adhesive climbers: These climbers achieve climbing by adhering to supports using adventitious roots or haustoria. Climbing typically occurs as long as the supporting surface provides adequate surface area, such as *Parthenocissus tricuspidata* (Sieb. & Zucc.) Planch. (Figure 4d).

The steps for identifying the climbing methods of climbing plants in this study are as follows: First, all species with tendrils were classified as tendrillar climbers. Species without tendrils were further classified into the other three climbing methods. The climbers with adventitious roots or haustoria were classified as adhesive climbers; those without the above specialized organs but with obvious twining behavior were referred to as twining climbers; and the remaining were classified as sprawling climbers. In addition, following the classification of Dias et al. [91], twining and tendrillar climbers were merged into the type of active climbing, and adhesive and sprawling climbers into passive climbing.

The final checklist of climbing plants in China mainly includes the following information: taxonomic status (family, genus, species, and infraspecific taxa), endemicity, life form, climbing method, endangered category, invasiveness, and geographical distribution across 28 provinces. Furthermore, the checklist primarily encompassed wild plant species in China, although it involved commonly cultivated species.

### 4.3. Environmental Variable Data

To evaluate the impacts of different environmental variables on the diversity patterns of climbing plants, we selected a total of 23 environmental variables from four categories (Table 6). These include the following: (1) Temperature variables, encompassing temperature-related data from Bio1 to Bio11. (2) Precipitation variables, including precipitation-related data from Bio12 to Bio19. The data on temperature and precipitation variables were sourced from the Worldclim database (http://www.worldclim.org/, accessed on 5 August 2025) with a resolution of 1 km^2^. (3) Geographical location variables, comprising the mean elevation and the midpoint values of longitude and latitude for the 28 provinces. The elevation data were obtained from the Resource and Environment Science Data Platform of the Chinese Academy of Sciences (http://www.resdc.cn/Default.aspx, accessed on 16 August 2025) with a resolution of 1 km^2^. (4) Human impact variable. The Human Footprint Map illustrates the impact of humans on each terrestrial biome globally, with values ranging from 0 (minimum impact) to 100 (maximum impact) (https://www.earthdata.nasa.gov/, accessed on 20 August 2025). To analyze the impact of human activities on the 28 provinces, the average anthropogenic influence for each province was calculated based on the Human Footprint data [84].

Using the function of Zonal Statistics in the Spatial Analyst of ArcMap (Environmental Systems Research Institute (ESRI). ArcGIS Desktop: ArcMap (Version 10.8), 2022, https://www.esri.com/, accessed on 6 July 2025), with a vectorized map of China’s provincial administrative units serving as the zonal layer for statistical analysis, we conducted statistical calculations on the raster data of environmental variables within each provincial unit. This allowed us to obtain the values of environmental variables for each provincial unit [92].

### 4.4. Data Analyses

To measure the diversity of climbing plant species at the family and genus levels in China, we employed the F-index and G-index. The F-index reflects diversity at the family level, and the G-index at the genus level. Higher values of the F-index and G-index indicate greater diversity at the family and genus levels, respectively. The calculation methods are as follows [93,94]:

(1) F-index, *D_F_*:

In a specific family *k*, 



$$D_{\mathrm{Fk}}=-\sum_{i=1}^{n}p_{i}{lnp}_{i}$$



Among them: *p_i_* = *s_ki_/S_k_*, where *s_ki_* = the number of species in genus *i*, *S_k_* = the total number of species in family *k*, *n* = the number of genera in family *k*.

The F-index of a region: *D_F_* = $\sum_{k=1}^{m}D_{Fk}$

Among them: *m* = the total number of families in the class.

(2) G-index, *D_G_*:



$$D_{G}=-\sum_{j=1}^{p}D_{Gi}=-\sum_{j=1}^{p}q_{j}{lnq}_{j}$$



Among them: *q_j_* = *s_j_*/*S*, *s_j_* = the number of species in genus *j* of a certain class, *S* = the total number of species in a certain class, *p* = the number of genera in a certain class.

Taking into account the effect of regional area on species richness and the variation in the area of each provincial unit, we adopted the climbing plants’ density index (*D-value*) to eliminate the effect of area differences among geographical units on species richness [95]. The *D-value* represents the area-corrected species richness of climbing plants. For each geographical unit, *D = N/log(A)*, where *N* is the number of climbing plant species in each provincial unit and *A* is its area. In this study, when analyzing the relationship between climbing plant species richness and environmental variables, we eliminated the effect of area on the distribution of climbing plants based on the *D-value*, and the species richness used in our analysis was actually species density. Furthermore, we distinguished between wild and cultivated species, and when counting the number of climbing plant species in each of the 28 geographical units, only wild species were accordingly included.

We examined the relationships between climbing plant species density in each provincial unit and the densities of endemic, threatened, and invasive climbing plants in China, and conducted correlation analyses between species density and environmental variables, using Pearson correlation coefficient analysis. The criteria for determining correlation coefficients are as follows: strong correlation for ∣*r*∣ > 0.66, moderate correlation for 0.66 ≥ ∣*r*∣ > 0.33, and weak correlation for ∣*r*∣ ≤ 0.33 [96]. In addition, we compared the correlations between climber species density and the densities of endemic, threatened, and invasive species in China using One-Way ANOVA. The statistics analysis was performed using Statistica 7 software (TIBCO Software Inc., Palo Alto, CA, USA). Considering that climbing plants of different life forms may respond differently to the selected environmental variables, we separately analyzed the correlations between the density of each life form (i.e., evergreen woody lianas, deciduous woody lianas, and herbaceous vines) and environmental variables.

Variation partitioning was employed to reveal the pure and shared effects of four explanatory variable groups (i.e., temperature variables, precipitation variables, geographical location variables, and human impact variable) on climbing plant species density. Variation partitioning analysis was conducted using the varpart function in the R package (vegan: Community Ecology Package; version 2.4-1, 2016), and results were presented based on adjusted R^2^ values. Additionally, variation partitioning was also performed for each life form. Given that variation partitioning is sensitive to the correlation and collinearity among the selected environmental variables, we first conducted cluster analysis on 19 climatic variables to eliminate the effects of multicollinearity [92]. Then, within each collinearity group, a variable was selected as the representative of that group based on the magnitude of the Pearson correlation coefficient for the final variation partitioning.

## 5. Conclusions

We first collated the most comprehensive and updated checklist of climbers in China involving 28 geographical units. Our results indicate that Chinese climbing plants (3485 species, 551 genera, 105 families) have a high level of diversity, marked by rich species, high endemism, various life forms, and diverse climbing methods. These climbers exhibit a spatially heterogeneous pattern, with species richness forming a decreasing gradient from south to north in China. Moreover, variation partitioning indicates that the pure effect of precipitation explained most of the variation in entire climber density (X2|X1 + X3 + X4, R^2^ = 0.44), and the same was found for the three different life forms (i.e., evergreen woody lianas, deciduous woody lianas, and herbaceous vines; X2|X1 + X3 + X4, R^2^ = 0.50, 0.45, 0.49). In conclusion, this study explicitly examines the diversity and spatial distribution of climbing plants across China and determines their key environmental drivers; it may accordingly inform other regional-scale climbing plant studies in the near future. Nonetheless, we acknowledge certain limitations in the data source, underscoring the necessity of integrating taxonomically verified data with time-calibrated molecular phylogenies for finer-grained insights into the spatial pattern of Chinese climbers. Furthermore, incorporating additional environmental variables—such as soil properties and land cover—could provide a more comprehensive understanding of climber species diversity and distribution pattern.

## Figures and Tables

**Figure 1 plants-14-03281-f001:**
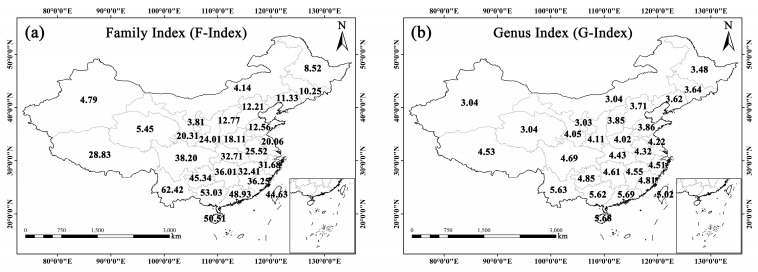
Spatial patterns of F-Index (**a**) and G-Index (**b**) for climbing plants at the provincial scale across China.

**Figure 2 plants-14-03281-f002:**
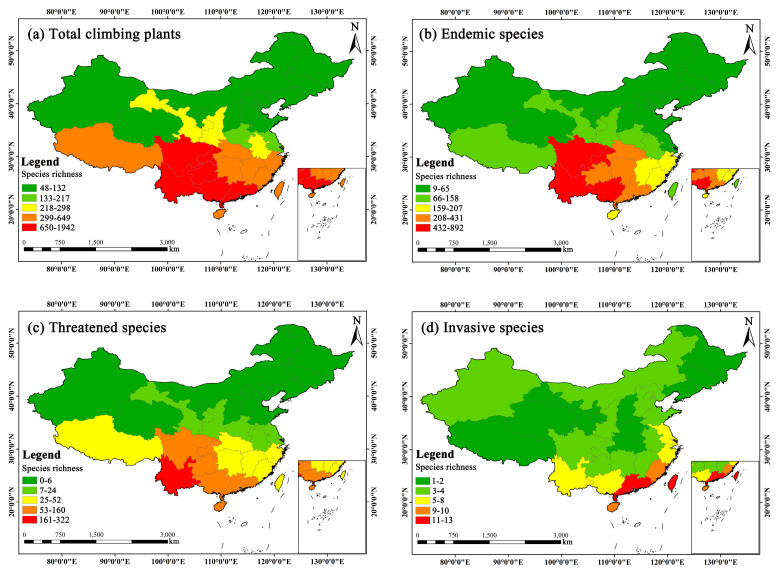
Spatial patterns of total (**a**), endemic (**b**), threatened (**c**), and invasive (**d**) climbing plants at the provincial scale across China.

**Figure 3 plants-14-03281-f003:**
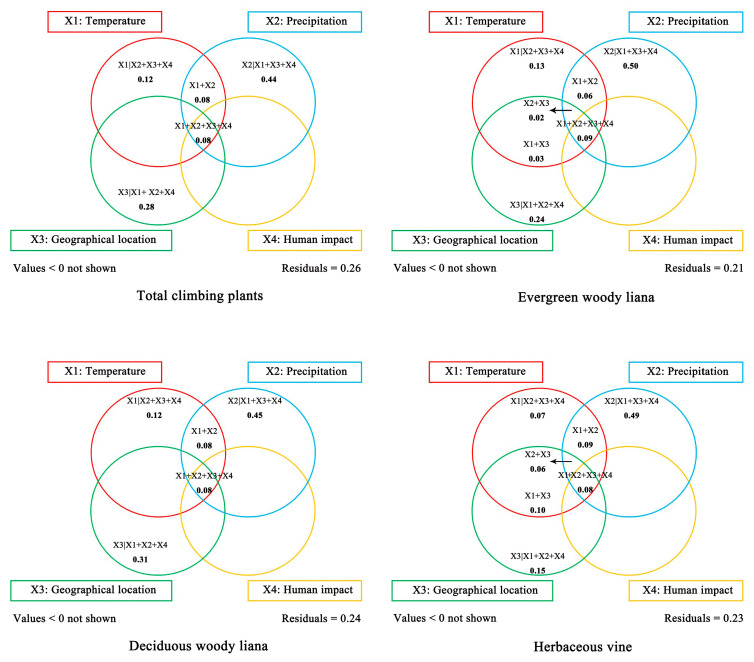
Variation partitioning for climbing plant species density across China. X1|X2 + X3 + X4: the pure effect of temperature; X2|X1 + X3 + X4: the pure effect of precipitation; X3|X1 + X2 + X4: the pure effect of geographical location; X4|X1 + X2 + X3: the pure effect of human impact.

**Figure 4 plants-14-03281-f004:**
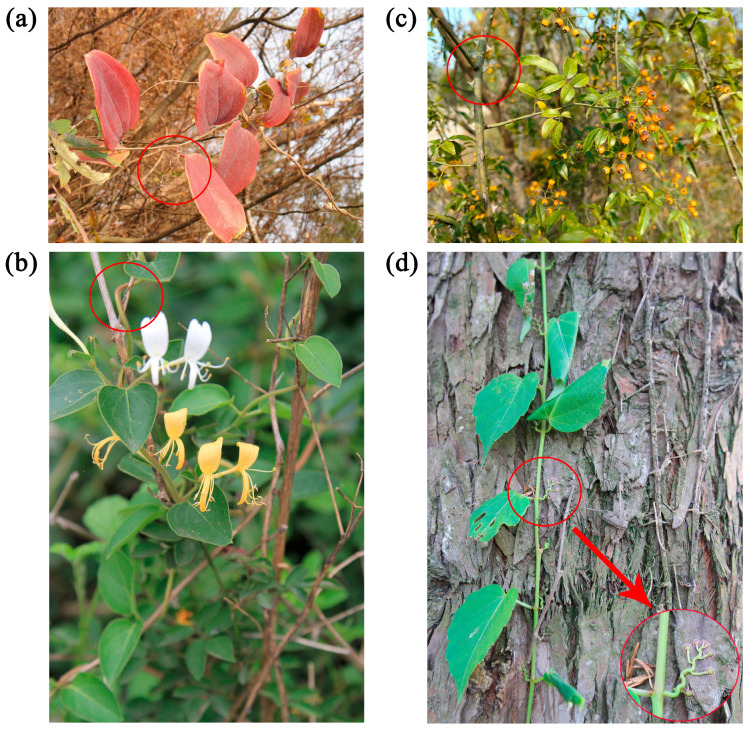
Photos of the four climbing methods of climbing plants. (**a**) Tendrillar climbers: *Smilax china* L.; (**b**) Twining climbers: *Lonicera japonica* Thunb.; (**c**) Sprawling climbers: *Rosa cymosa* Tratt.; (**d**) Adhesive climbers: *Parthenocissus tricuspidata* (Sieb. & Zucc.) Planch. A red circle in the bottom-right corner denotes the swollen suckers at the tentril apex. The photos were taken by Guangfu Zhang.

**Table 1 plants-14-03281-t001:** Taxa and species composition of Chinese climbing plants.

Taxa	Family		Genus		Species	
	Number	Percentage	Number	Percentage	Number	Percentage
Lycophytes and Ferns	3	2.86%	3	0.54%	11	0.32%
Gymnosperms	1	0.95%	1	0.18%	10	0.29%
Angiosperms	101	96.19%	547	99.28%	3464	99.39%
Total	105	100.00%	551	100.00%	3485	100.00%

**Table 2 plants-14-03281-t002:** Family groups of Chinese climbing plants. “*N*” represents the number of species, and “Number” represents the counts of family, genus and species in four different types of families, respectively.

Types	Family		Genus		Species	
	Number	Percentage	Number	Percentage	Number	Percentage
Large family (*N* > 100)	9	8.57%	242	43.92%	1893	54.32%
Medium family (6 ≤ *N* < 100)	56	53.33%	254	46.10%	1515	43.47%
Oligospecific family (2 ≤ *N* ≤ 5)	19	18.10%	34	6.17%	56	1.61%
Monotypic family (*N* = 1)	21	20.00%	21	3.81%	21	0.60%
Total	105	100.00%	551	100.00%	3485	100.00%

**Table 3 plants-14-03281-t003:** Life form and climbing method of Chinese climbing plants.

Life Form	Climbing Method				Total (%)
	Active		Passive		
	Twining Climbers	Tendrillar Climbers	Adhesive Climbers	Sprawling Climbers	
Evergreen woody liana	712	148	129	373	1362 (39.08%)
Deciduous woody liana	431	148	39	276	894 (25.65%)
Herbaceous vine	686	269	46	228	1229 (35.27%)
Total	1829 (52.48%)	565 (16.21%)	214 (6.14%)	877 (25.17%)	3485 (100.00%)

**Table 4 plants-14-03281-t004:** Correlation among climbing plants’ species density with endemic, threatened and invasive species density. Note: Endemic, threatened and invasive in the table all refer to the corrected species density. For more details, see the text. Note: ** represents significant difference (*p* < 0.01).

	Species Density	Endemic	Threatened	Invasive
Species density	1			
Endemic	0.950 **	1		
Threatened	0.963 **	0.912 **	1	
Invasive	0.711 **	0.499 **	0.617 **	1

**Table 5 plants-14-03281-t005:** Pearson’s correlation coefficients between species density and environmental variables. Note: Climbing plants, evergreen woody liana, deciduous woody liana and herbaceous vine in the table all refer to the corrected species density.

Variables	Climbing Plants	Evergreen Woody Liana	Deciduous Woody Liana	Herbaceous Vine
Bio1	0.722	0.695	0.755	0.681
Bio2	−0.696	−0.647	−0.748	−0.676
Bio3	0.610	0.620	0.496	0.629
Bio4	−0.862	−0.822	−0.863	−0.854
Bio5	0.216	0.216	0.272	0.158
Bio6	0.815	0.778	0.846	0.783
Bio7	−0.876	−0.831	−0.887	−0.864
Bio8	0.352	0.347	0.359	0.326
Bio9	0.821	0.787	0.847	0.787
Bio10	0.365	0.353	0.417	0.316
Bio11	0.828	0.793	0.855	0.796
Bio12	0.754	0.699	0.766	0.767
Bio13	0.709	0.652	0.694	0.749
Bio14	0.565	0.505	0.626	0.566
Bio15	−0.511	−0.455	−0.650	−0.452
Bio16	0.766	0.713	0.747	0.796
Bio17	0.546	0.486	0.617	0.540
Bio18	0.760	0.701	0.728	0.812
Bio19	0.535	0.478	0.600	0.530
Latitude	−0.864	−0.835	−0.886	−0.822
Longitude	−0.049	−0.082	−0.032	−0.007
Elevation	−0.198	−0.174	−0.252	−0.178
Human impact	0.227	0.162	0.300	0.255

**Table 6 plants-14-03281-t006:** The classification and representation of environmental variables used in this study.

Category	Variable	Description	Unit
Temperature	Bio1	Annual mean temperature	°C
	Bio2	Mean diurnal range (mean of monthly (max temp–min temp))	°C
	Bio3	Isothermality ((Bio2/Bio7) × 100)	%
	Bio4	Temperature seasonality (standard deviation × 100)	-
	Bio5	Max temperature of warmest month	°C
	Bio6	Min temperature of coldest month	°C
	Bio7	Temperature annual range (Bio5–Bio6)	°C
	Bio8	Mean temperature of wettest quarter	°C
	Bio9	Mean temperature of driest quarter	°C
	Bio10	Mean temperature of warmest quarter	°C
	Bio11	Mean temperature of coldest quarter	°C
Precipitation	Bio12	Annual precipitation	mm
	Bio13	Precipitation of wettest month	mm
	Bio14	Precipitation of driest month	mm
	Bio15	Precipitation seasonality (coefficient of variation)	-
	Bio16	Precipitation of wettest quarter	mm
	Bio17	Precipitation of driest quarter	mm
	Bio18	Precipitation of warmest quarter	mm
	Bio19	Precipitation of coldest quarter	mm
Geographical location	Latitude	midpoint values of latitude	°
	Longitude	midpoint values of longitude	°
	Elevation	Average elevation	m
Human impact	HI	Human footprint	-

## Data Availability

The original contributions presented in this study are included in the article/Appendix A. Further inquiries can be directed to the corresponding author.

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
