# Peer review of "Diversity, Pattern, and Environmental Drivers of Climbing Plants in China"

_plants, 2025, doi:10.3390/plants14213281_

Round 1

Reviewer 1 Report

Comments and Suggestions for Authors

This Ms. studied diversity distribution patterns and environmental drivers of climbing plants in China, covering 3485 species, and found woody lianas dominated the climbing flora, and also found Chinese climbers largely presented a pattern of species richness decreasing from south to north, etc. there are some important findings which are important for us to understand further conservation of climbing plants. 

However, there are also several points that need further improvement:

The introduction may be too long, should be abbreviated to focus on the topic clearly.

Environmental determinants are chosen as key words, only emerging two times in the text, please reconsider.

CPC is confusion, there is no full name.

3.1, the discussion part should be reorganized to focus on the topic.

The sentence in the abstract, “This study also offers a framework 25 on climbers at regional scales for future studies”, seems exaggerated its significance and should be recognized.

Comments on the Quality of English Language

Mostly the language is good, but several parts need further improvement.

Author Response

To Reviewer 1

General comments:

"This Ms. studied diversity distribution patterns and environmental drivers of climbing plants in China, covering 3485 species, and found woody lianas dominated the climbing flora, and also found Chinese climbers largely presented a pattern of species richness decreasing from south to north, etc. there are some important findings which are important for us to understand further conservation of climbing plants. However, there are also several points that need further improvement."

√ We firstly express gratitude to the first reviewer for his/her critical review. Then, we reply to his/her specific comments point by point.

Specific comments:

  1. " The introduction may be too long, should be abbreviated to focus on the topic clearly. "

√ Thank you very much for your comment. We have accepted your suggestion, and made revisions in the section of Introduction. Namely, we have merged the third, fourth and fifth paragraphs in the original text into one paragraph.

Please see Line 83-96 for more details.

  1. " Environmental determinants are chosen as key words, only emerging two times in the text, please reconsider. "

√ Thank you very much for your comment.

We have accepted the suggestion, and replaced "environmental determinants" with "environmental variables" in the section of Keywords.

Please see Line 27-28.

  1. " CPC is confusion, there is no full name. "

√ Thank you very much for your comment. According to the suggestion of the Reviewer, we have made a revision in the text.

The full name of CPC is "climbing plants in China". We have defined the acronym "CPC" upon its first use in the manuscript.

Please see Line 206.

  1. " 3.1, the discussion part should be reorganized to focus on the topic. "

√ Thanks for your critical review. We have accepted your suggestion, and reorganized the contents in the section of Discussion (i.e., 3.1 Diversity Characteristics of Climbing Plants in China).

Please see Line 389-391, 399-402, 409-410, 425-428, 437 for more details.

  1. "The sentence in the abstract, “This study also offers a framework on climbers at regional scales for future studies”, seems exaggerated its significance and should be recognized."

√ Thank you for your comment. We have accepted the suggestion, and made a revision herein.

The original sentence has been changed into the following one:

"This study also provides a valuable reference on climbers at regional scales for future studies."

Furthermore, we have checked the whole Manuscript, and made corresponding revisions in the sections of Introduction and Discussion.

Please see Line 24-25, 207, 553 for more details.

  1. " Comments on the Quality of English Language: Mostly the language is good, but several parts need further improvement."

√ Thank you for your comment. According to the suggestion of the Reviewer, we have carefully checked and polished the whole text. For more details, please see the revised Manuscript.

Reviewer 2 Report

Comments and Suggestions for Authors

After reviewing the manuscript entitled "Diversity, Pattern, and Environmental Drivers of Climbing Plants in China," I comment as follows:

The work very clearly and precisely compiles the taxonomic diversity of climbing plants in China. It provides a very detailed analysis of how diversity is distributed among different families and the environmental factors that influence species distribution patterns.

In my opinion, and subject to the editor-in-chief's discretion, I suggest its acceptance for publication.

Regards

Author Response

To Reviewer 2

General comments:

" After reviewing the manuscript entitled "Diversity, Pattern, and Environmental Drivers of Climbing Plants in China," I comment as follows:

The work very clearly and precisely compiles the taxonomic diversity of climbing plants in China. It provides a very detailed analysis of how diversity is distributed among different families and the environmental factors that influence species distribution patterns. In my opinion, and subject to the editor-in-chief's discretion, I suggest its acceptance for publication.

Regards"

√ We express gratitude to the second reviewer for his/her critical review and encouraging comments.

Reviewer 3 Report

Comments and Suggestions for Authors

Comments and Suggestions for the Authors

To the Authors,

The manuscript entitled “Diversity, Pattern, and Environmental Drivers of Climbing Plants in China” by Wang & Zhang provides a comprehensive analysis of the botanical contingent of climbing species in China. It presents a spatial assessment that considers environmental factors as the main drivers influencing their occurrence, distribution, and diversity.

The paper is complete and of considerable importance within the global botanical framework. It presents some results that could potentially improve the management of climbing plant species, particularly those that are endemic or threatened. Therefore, the authors are encouraged to carefully consider the feedback provided below and revise the manuscript accordingly:

L11: It would be preferable to use “collected” rather than “collated”, as it is more direct in this context (Abstract).

L32: References should follow the numerical format [1], [2], etc. Please adjust accordingly.

L195: Acronyms (e.g., CPC) should be defined upon first use in the text.

L201: The use of the % symbol could improve the clarity of Table 1.

L222: Similarly, the % symbol could be used in Table 2 to enhance clarity; it would also be advisable to specify in the caption what “N” and “Number” refer to, to avoid confusion.

L262: The phrase “familial diversity decreases from south to north” does not require quotation marks.

L265: Figures should be enlarged for better readability. In addition, it would be useful to specify Family Index (F-Index) and Genus Index (G-Index) within the figure.

L282: In “species richness decreasing from south to north”, quotation marks can be removed.

L300, L448: Remove quotation marks as above.

L629: When introducing a species for the first time, it is important to provide the full scientific name including the author citation, e.g. Smilax china L. This should be applied consistently throughout the manuscript.

L642: The letters [(a), (b), etc.] in the figures are difficult to read; a white background or higher contrast would improve legibility.

L667: The resolution should also be expressed in km² for comparability.

L679: Please include the full citation of the software used.

Additional Comments

Materials and Methods

Selection of environmental variables:
At such a broad spatial scale, and given that several climbing species are closely linked to soil composition and canopy structure—particularly in forested habitats—it would be valuable to include additional environmental variables related to soil (e.g., texture, % rock, sand, silt and clay, pH, macro- and micronutrient content, moisture, cation exchange capacity, etc.) and land cover (e.g. different type of forest habitats that could be associated with climbing species).
Incorporating these parameters, if available, would provide a more comprehensive understanding of species distribution and diversity patterns.

Discussion

I recommend further elaboration and clarification in the “Discussion” section, particularly taking into account the results explained in the paragraph “2.5. Correlations among the Entire, Endemic, Threatened, and Invasive Climbing Plant Species Density in China” and the corresponding Table 4 from the “Results” section
The interpretation of this part is currently unclear, especially regarding the meaning of the reported results and the correlations among the different groups. It is crucial to specify whether the observed correlations are statistically significant or not, and to discuss their potential ecological implications.

Author Response

To Reviewer 3

General comments:

" Comments and Suggestions for the Authors

To the Authors,

The manuscript entitled “Diversity, Pattern, and Environmental Drivers of Climbing Plants in China” by Wang & Zhang provides a comprehensive analysis of the botanical contingent of climbing species in China. It presents a spatial assessment that considers environmental factors as the main drivers influencing their occurrence, distribution, and diversity.

The paper is complete and of considerable importance within the global botanical framework. It presents some results that could potentially improve the management of climbing plant species, particularly those that are endemic or threatened. Therefore, the authors are encouraged to carefully consider the feedback provided below and revise the manuscript accordingly."

√ We appreciate the third reviewer for his/her critical review. Then, we reply to his/her specific comments point by point. Please see the following response and revised Manuscript.

Specific comments:

  1. " L11: It would be preferable to use “collected” rather than “collated”, as it is more direct in this context (Abstract). "

√ Thank you for your comment. We have accepted the suggestion, and made a revision herein.

Please see Line 11-12.

  1. " L32: References should follow the numerical format [1], [2], etc. Please adjust accordingly."

√ Thank you very much for your comment. We have accepted your suggestion and made revisions in the manuscript. For more details, please see the citations in the text and the reference list.

  1. " L195: Acronyms (e.g., CPC) should be defined upon first use in the text."

√ Thank you for your comment. We have accepted the suggestion and defined the acronym "CPC" upon its first use in the manuscript.

Please see Line 206 for more details.

  1. " L201: The use of the % symbol could improve the clarity of Table 1."

√ Thanks for your comment. We have accepted the suggestion and added the percentage symbol (i.e., %) in Table 1.

Please see Line 220.

  1. " L222: Similarly, the % symbol could be used in Table 2 to enhance clarity; it would also be advisable to specify in the caption what “N” and “Number” refer to, to avoid confusion."

√ Thank you for your valuable comments. We have accepted your suggestions, and made revisions in Table 2.

Please see Line 237-240.

  1. " L262: The phrase “familial diversity decreases from south to north” does not require quotation marks."

√ Thanks for your comment. We have accepted your suggestion and deleted quotation marks. In addition, we have checked the whole text and made the corresponding revisions.

  1. " L265: Figures should be enlarged for better readability. In addition, it would be useful to specify Family Index (F-Index) and Genus Index (G-Index) within the figure."

√ Thank you very much for your comment. We have accepted your suggestion and made modifications in Figure 1.

Please see Line 281.

  1. " L282: In “species richness decreasing from south to north”, quotation marks can be removed."

√ Thank you for your comment. We have accepted your suggestion and removed quotation marks.

  1. " L300, L448: Remove quotation marks as above."

√ Thanks for your comment. We have accepted your suggestion and deleted quotation marks.

  1. " L629: When introducing a species for the first time, it is important to provide the full scientific name including the author citation, e.g. Smilax china L. This should be applied consistently throughout the manuscript."

√ Thank you very much for your comment. We have accepted the suggestion and provided the full scientific name for each plant species throughout the manuscript.

Please see Line 183-184, 678, 683, 687, 690 for details.

  1. " L642: The letters [(a), (b), etc.] in the figures are difficult to read; a white background or higher contrast would improve legibility."

√ Thank you for your comment. We have accepted the suggestion and made modifications in Figure 4. For more details, please see the revised Figure 4.

Please see Line 691 for details.

  1. " L667: The resolution should also be expressed in km² for comparability."

√ Thank you very much for your comment. We have accepted your suggestion and made a revision herein.

Please see Line 717.

  1. " L679: Please include the full citation of the software used."

√ Thanks for your comment. We have accepted your suggestion and made a modification herein.

Please see Line 729-731.

  1. " Additional Comments

Materials and Methods

Selection of environmental variables:

At such a broad spatial scale, and given that several climbing species are closely linked to soil composition and canopy structure—particularly in forested habitats—it would be valuable to include additional environmental variables related to soil (e.g., texture, % rock, sand, silt and clay, pH, macro- and micronutrient content, moisture, cation exchange capacity, etc.) and land cover (e.g. different type of forest habitats that could be associated with climbing species).

Incorporating these parameters, if available, would provide a more comprehensive understanding of species distribution and diversity patterns. "

√ Thank you very much for your comments.

Regarding the environmental variables in this study, here is our understanding.

First, at the regional scale, plant distribution is primarily constrained by climate conditions (i.e., temperature and precipitation) (Huang et al., 2021). In contrast, soil exerts more influences at the local scale than climate.

Second, China spans 9.6 ×106 km², making it the world's third-largest country in territory. It extends 5,200 km from east to west, covering 62 degrees of longitude and 50 degrees of latitude, with five climatic zones from south to north: tropical, subtropical, warm temperate, temperate, and cold temperate. Correspondingly, vegetation types exhibit distinct latitudinal zonation—ranging from tropical rainforests, evergreen broadleaved forests, deciduous broadleaved forests, mixed coniferous-broadleaved forests, to coniferous forests. Similarly, soil properties also display evident zonal characteristics from south to north in China, while variations with elevation occur within the same climatic region (Zhang et al., 2022). This is for the first time to examine the distribution pattern and environmental drivers of climbers in China, therefore we selected 23 environmental variables across four categories (i.e., temperature, precipitation, geographical location, and human impact), based on existing literature and our understanding concerning climber distribution. We tend to think that the geographical variables including longitude, latitude, and elevation to a great extent indirectly reflect zonal soil types and altitudinal soil variations.

Third, land cover generally includes forests, shrublands, grasslands, barren, farmland, urban and water. Previous studies indicate that future climate change will affect the distribution of woody plants in China resulting from land use change (Peng et al., 2022). Climbers, due to their unique characteristics, can occur across various habitats—such as abandoned fields, croplands, urban peripheries, grasslands, shrublands, and forests—under temperate, subtropical, and tropical climates. Unlike woody plants (i.e. trees or shrubs), climbers are able to occur under various habitat conditions like forests, shrublands, grasslands, or even barren land with the same climate. In other words, unlike woody plants, it seems that there are not strong relationships between climbers and land cover in geographical distribution. Additionally, in the current study we used the variable "Human Impact", which may reflect land cover to some extent across different provinces.

As stated above, following the reviewer's suggestions, we have supplemented and revised the limitations of environmental variables in the section of Conclusions. For the time being, we do not add other environmental variables in this study.

Please see Line 804-807.

References:

Peng, S.J.; Zhang, J.; Zhang, X.L.; Li, Y.Q.; Liu, Y.P.; Wang, Z.H. Conservation of Woody Species in China under Future Climate and Land‐Cover Changes. J. Appl. Ecol. 2022, 59, 141-152.

Huang, E.H.; Chen, Y.X.; Fang, M.; Zheng, Y.; Yu, S.X. Environmental Drivers of Plant Distributions at Global and Regional Scales. Glob. Ecol. Biogeogr. 2021, 30, 697-709.

Zhang, J.H.; Zhu, L.Q.; Li, G.D.; Zhao, F.; Qin, J.T. Distribution Patterns of SOC/TN Content and Their Relationship with Topography, Vegetation and Climatic Factors in China's North-South Transitional Zone. J. Geogr. Sci. 2022, 32, 645-662.

  1. " Additional Comments

Discussion

I recommend further elaboration and clarification in the “Discussion” section, particularly taking into account the results explained in the paragraph “2.5. Correlations among the Entire, Endemic, Threatened, and Invasive Climbing Plant Species Density in China” and the corresponding Table 4 from the “Results” section.

The interpretation of this part is currently unclear, especially regarding the meaning of the reported results and the correlations among the different groups. It is crucial to specify whether the observed correlations are statistically significant or not, and to discuss their potential ecological implications."

√ Thanks for your critical comments. According to the suggestion of the Reviewer, we have conducted statistical analysis and made corresponding revisions.

Firstly, we have added statistical analysis for the correlation coefficients among the four groups (i.e., Entire, Endemic, Threatened, and Invasive climbers). Based on the results of significance tests, the differences among the four categories of climbing plant species were all found to be highly significant. Accordingly, we have revised Table 4 by modifying its caption and adding significance indicators. In addition, we have supplemented a statistic description approach to examine whether these correlations differed significantly or not in the section of Materials and Methods.

Secondly, we elaborate on the correlations observed across the different categories of climbing plants and state their potential ecological implications in the section of Discussion. For more details, please see the section of 3.2 Spatial Distribution Pattern of Climbers in China in the revised text.

Please see Line 445-464, 770-773 for more details.

Round 2

Reviewer 3 Report

Comments and Suggestions for Authors

Dear Authors,

Thank you for carefully addressing my comments in the revised version of the manuscript. I believe the paper has been significantly improved at this stage.

Congratulations on your work.

Kind regards